# SoMi-ToM: Evaluating Multi-Perspective Theory of Mind in Embodied Social Interactions

**Xianzhe Fan**
The University of Hong Kong
xianzhef@connect.hku.hk

**Xuhui Zhou**
Carnegie Mellon University

**Chuanyang Jin**
Johns Hopkins University

**Kolby Nottingham**
University of California Irvine

**Hao Zhu**
Stanford University

**Maarten Sap**
Carnegie Mellon University

## Abstract

Humans continuously infer the states, goals, and behaviors of others by perceiving their surroundings in dynamic, real-world social interactions. However, most Theory of Mind (ToM) benchmarks only evaluate static, text-based scenarios, which have a significant gap compared to real interactions. We propose the SoMi-ToM benchmark, designed to evaluate multi-perspective ToM in embodied multi-agent complex social interactions. This benchmark is based on rich multimodal interaction data generated by the interaction environment SoMi, covering diverse crafting goals and social relationships. Our framework supports multi-level evaluation: (1) first-person evaluation provides multimodal (visual, dialogue, action, etc.) input from a first-person perspective during a task for real-time state inference, (2) third-person evaluation provides complete third-person perspective video and text records after a task for goal and behavior inference. This evaluation method allows for a more comprehensive examination of a model's ToM capabilities from both the subjective immediate experience and the objective global observation. We constructed a challenging dataset containing 35 third-person perspective videos, 363 first-person perspective images, and 1225 expert-annotated multiple-choice questions (three options). On this dataset, we systematically evaluated the performance of human subjects and several state-of-the-art large vision-language models (LVLMs). The results show that LVLMs perform significantly worse than humans on SoMi-ToM: the average accuracy gap between humans and models is 40.1% in first-person evaluation and 26.4% in third-person evaluation. This indicates that future LVLMs need to further improve their ToM capabilities in embodied, complex social interactions.

 **Benchmark & Code:** github.com/XianzheFan/SoMi-ToM
 **Data & Dataset Card:** huggingface.co/datasets/SoMi-ToM/SoMi-ToM

## 1 Introduction

As AI systems increasingly interact with humans in complex environments, possessing Theory of Mind (ToM) capabilities [33]—the ability to understand mental states such as beliefs, goals, behaviors, attitudes, knowledge, desires, and emotions—is becoming increasingly important. Some research suggests [2, 36] that ToM not only helps agents understand the internal states of others but also assists them in efficiently executing tasks or interacting with others, especially in **embodied multi-party**

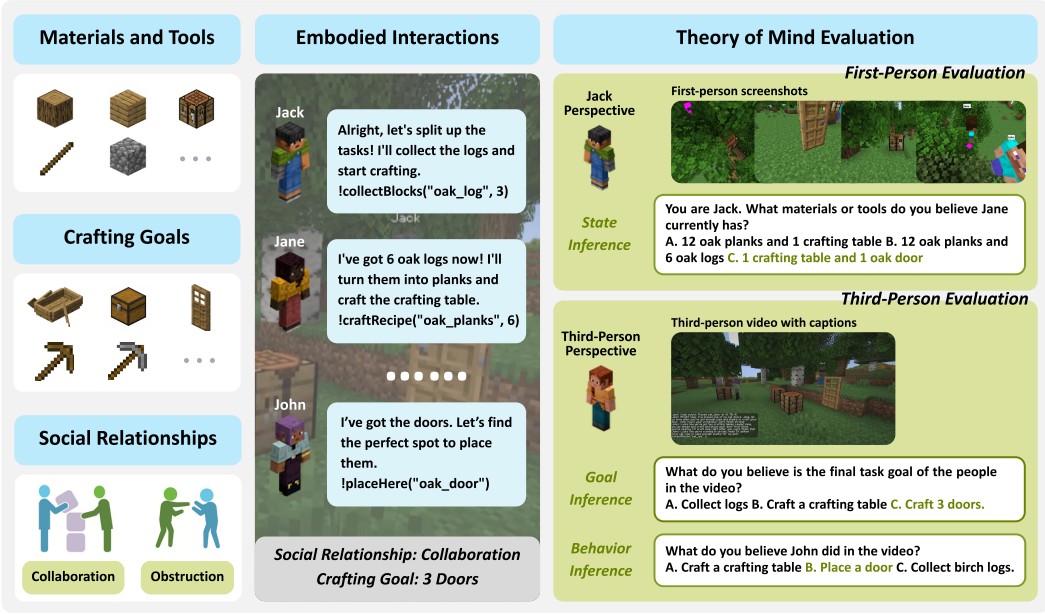

Figure 1: SoMi-ToM is built upon data from embodied AI agent interactions in Minecraft. There are a total of 35 tasks, and each task assigns a crafting goal that the agent needs to achieve by collecting materials and crafting tools. Simultaneously, there are different social relationships between agents: *collaboration* or *obstruction*. The evaluation is multi-perspective: (1) First-person (1050 questions): inferring the *state* during real-time interaction. (2) Third-person (175 questions): inferring the *goal* and *behavior* based on the complete video. In the examples, the green option is the correct answer.

**interaction scenarios**. In these situations, the operation of ToM often relies on the integration of multimodal information [2, 23, 36, 41]. This integration allows agents to connect observed behaviors with the surrounding environment and potential goals, in order to accurately infer the mental states of various parties and thus achieve effective collaboration or obstructive strategies. However, robustly measuring ToM for embodied settings remains a challenge [36].

We developed the SoMi embodied interaction environment, which supports LVLM agents controlling characters in the open-world game Minecraft and interacting with other agents. SoMi not only covers diverse crafting goals (e.g., craft a chest) but also incorporates complex social dynamics, including cooperating to complete tasks or hindering (e.g., using irrelevant speech to delay other team members in crafting the chest). Based on the rich multimodal interaction data generated by SoMi, we introduce SoMi-ToM, a novel benchmark designed to evaluate the **ToM** of models in complex **so**cial interactions among embodied agents in **Mi**necraft. SoMi-ToM addresses the shortcomings of current ToM benchmarks: (1) Lack of evaluation of embodied agents' ToM when interacting with the environment and others, especially in complex tasks and open-world scenarios [23, 36]. (2) Failure to adequately explore diverse social relationships and interactions among multiple agents [7, 25]. (3) Failure to evaluate ToM based on information from different perspectives and modalities. Existing ToM datasets typically rely only on visual or textual [10, 15, 26, 34, 38, 46] input from a single perspective (third-person [17, 48] or first-person [21, 41]).

SoMi-ToM consists of multi-perspective, multi-stage, and multimodal question-answering (all are multiple-choice questions) based on 35 tasks where AI agents accomplish various goals (Figure 1). During task execution, we combine **first-person perspective** multimodal inputs (including game screenshots, multi-agent dialogue transcripts, actions, game feedback, and rule introductions) to real-time evaluate the agent's *state inference* ability regarding the resources held by itself and other agents. This ability is crucial for immediate planning and inter-agent interaction. After the tasks are completed, we utilize subtitled videos from a **third-person perspective** to evaluate the agent's *goal inference* and *behavior inference* – two ToM capabilities – aiming to examine the model's ability to understand complex interactions holistically.

We conducted a systematic evaluation of humans and current state-of-the-art Large Vision Language Models (LVLMs) on SOMI-TOM. The results indicate that LVLMs perform significantly below human level on SOMI-TOM. Specifically, in the first-person evaluation, the average accuracy gap between humans and LVLMs is 40.1%. Notably, most models showed improved accuracy after using the Chain-of-Thought (CoT) method. In the third-person evaluation, the average accuracy gap between humans and LVLMs is 26.4%. However, for most models except Gemini 2.0 Flash, GPT-4o, and Qwen2.5-VL, the average accuracy decreased after using CoT. Furthermore, LVLMs performed better on *goal inference* tasks than on *behavior inference* tasks from this perspective. Qualitative analysis indicates that the main reasons for the poor performance of LVLMs include: ignoring or inaccurately tracking resource consumption, insufficient reliance on system feedback, being misled by initial intentions rather than actual behavior, overgeneralization or inappropriate associations, failure to identify hierarchical goal structures, entity recognition confusion, and detailed errors.

In summary, our main contributions include: (1) Development of SOMI, an embodied multi-agent social interaction environment in Minecraft. (2) A novel embodied ToM benchmark, SOMI-TOM, for evaluating multi-perspective ToM in complex multi-agent social interactions within Minecraft. (3) A systematic evaluation of state-of-the-art LVLMs, and validation of the dataset through human experiments, providing human baseline performance.

## 2 Related Work

**Theory of Mind Benchmarks.** The assessment of a model's ToM capabilities often draws on tests historically used to evaluate human ToM abilities [3, 4, 19]. Single-agent ToM benchmarks [14, 16, 23, 26, 37] have extensively tested concepts such as beliefs, goals, preferences, constraints, and rationality. Multi-agent benchmarks are often based on the classic Sally-Anne test [3] and are used to test false beliefs and higher-order beliefs [27, 38, 46, 48]. Text-based multi-agent benchmarks like FANToM [25], ToMBench [10], NegotiationToM [7], EmoBench [34] and Li et al.'s work [29] test the beliefs and intentions of agents in complex dialogues or interactions but do not involve relationships between agents (such as collaboration or obstruction). Research on inter-agent relationship understanding and ToM, such as Phase [32] and the Infant Cognition Benchmark [30], relies on simple 2D animations and lacks embodied, human-like interactions. These ToM benchmarks primarily focus on single-modal inputs like text, images, or video, and thus cannot comprehensively evaluate a model's ToM capabilities in real-world environments.

**Multimodal Theory of Mind Benchmarks.** Multimodal ToM benchmarks primarily evaluate models' ability to integrate various modalities, including text, images, and video input, to infer mental states [50]. Groenestijn et al. proposed a benchmark that combines human behavior and inner monologues while performing object rearrangement tasks in a simulated environment with robot interaction, aiming to assess the LVLM's ability to reason about others' belief changes [41]. Thewes, in the Mindcraft multimodal environment [2], proposed an integrated model architecture that combines video, text, and knowledge graphs. This approach enhances LVLMs' common ground reasoning capabilities in human-robot collaboration tasks through implicit ToM reasoning [40]. Das et al., through iterative ToM tests based on image and text inputs, revealed the limitations of LVLMs in constructing unified world models and processing low-resource languages (such as Bengali) [12]. CHARTOM evaluates the LVLM's ability to understand charts and determine if a chart might mislead human readers [6]. WhodunitBench, based on murder mystery games (image and text input), is used to evaluate the performance of large-scale multimodal agents in areas such as compositional skills, multi-agent collaboration, and multi-step reasoning [47]. Chen et al. proposed exploratory questions for videos with rich social and emotional reasoning content, developed an LVLM pipeline for ToM reasoning using video and text, and achieved ToM reasoning by retrieving keyframes [8]. MMToM-QA evaluates ToM for single-agent behavior through multimodal inputs [23]. MuMA-ToM tests agents' social intentions and their reasoning about each other's mental states within the context of two agents interacting in an embodied home environment [36]. However, these multimodal ToM benchmarks, due to their single perspective, simple relationships between agents, simple task design, and lack of continuous evaluation of dynamic ToM during task execution, make it difficult to comprehensively measure models' abilities in real-time adaptation and understanding complex tasks and social situations. As shown in Table 7 (Appendix C), unlike previous work, SOMI-TOM is a novel benchmark for evaluating multi-perspective ToM in complex social interactions among embodied multiple agents. The videos in our dataset are also significantly longer (Table 6).

# 3  SoMi Embodied Interaction Environment

To evaluate ToM in embodied social interaction, we developed the SoMi interactive environment. This environment is easily extendable and supports LVLM agents controlling characters in the open-world game Minecraft, allowing them to collaborate with other agents to achieve crafting goals. The interaction logs, game screenshots, and videos generated by the interactive environment will be used for the SoMi-ToM evaluation. To develop this environment, we designed a three-tiered asynchronous communication architecture to enable autonomous multi-agent interaction without human intervention, addressing the limitation of existing frameworks that only support a single agent[1]. Furthermore, we endowed the agents with visual perception by integrating a first-person screenshot capability, and balanced the benchmark's reproducibility with interaction diversity by fixing the world seed and randomizing spawn points. Development details can be found in Appendix D.

## 3.1  Minecraft Game Episode Setup and Crafting Rules

SoMi could support the interaction among any number of agents, but we consider tasks that involve three LVLM agents in this paper. When agents enter Minecraft, different material blocks (e.g., log and cobblestone) will be distributed in the environment[2]. The agents will output **dialogue content** with other agents and **action commands**, completing the final task by collecting material blocks, crafting new materials and tools. When an agent interacts with the environment, the game provides corresponding **system feedback**, such as whether material collection, item crafting, or action execution was successful. This feedback is only visible to the agent itself and external observers, and is not known to other agents. The crafting process involves 10 types of materials or tools: log, plank, stick, crafting table, chest, boat, door, wooden pickaxe, cobblestone, stone pickaxe (Table 1).

---

**Example Dialogue Content & System Feedback**

John: Jane, since you've crafted the crafting table, go ahead and place it!
Jane: I'll place it now so we can move on. `!placeHere("crafting_table")`
Jane: Alright, crafting table is placed! Jack, go ahead and craft the sticks, then the wooden pickaxe. I'll mine stone as soon as it's ready.
Jack: Jane, since you've placed the crafting table, I'll craft the wooden pickaxe now. `!craftRecipe("wooden_pickaxe", 1)`
**system:** The status of Jack's action execution: Found crafting_table at (11, 70, 9). You have reached at 11, 70, 9. Successfully crafted wooden_pickaxe, you now have 1 wooden_pickaxe.

**Example Action Commands**

`!collectBlocks("birch_log", 5)`: The agent hits a specific block to collect it (e.g., birch log). This process can be repeated to collect multiple blocks of this type.
`!craftRecipe("crafting_table", 1)`: The agent crafts different material blocks into new blocks. For example, four planks can be crafted into a crafting table.
`!moveAway(10)`: Moves arbitrarily 10 units distance.
`!goToBlock("stone", 2, 10)`: Searches for the nearest stone block within a 10-unit radius and attempts to move to a location 2 units away from the stone block.
`!goToPlayer("John", 2)`: Moves to a location 2 units away from John.
`!givePlayer("John", "stick", 4)`: Gives John 4 sticks.
`!nearbyBlocks`: Queries the types and quantities of nearby material blocks.

---

Table 1: Rules for crafting tools and collecting materials in Minecraft.

| Task level | Task | Reasoning Steps | Recipe | Tool/Platform |
|---|---|---|---|---|
| Basic level | Mine logs | $\mathcal{O}_1$ | - | - |
| | Craft planks | $\mathcal{O}_2$ | 1*log (yields 4*planks) | - |
| | Craft sticks | $\mathcal{O}_3$ | 2*planks | - |
| | Craft crafting tables | $\mathcal{O}_3$ | 4*planks | - |
| Wooden level | Craft boats | $\mathcal{O}_4$ | 5*planks | crafting table |
| | Craft chests | $\mathcal{O}_4$ | 8*planks | crafting table |
| | Craft doors | $\mathcal{O}_4$ | 6*planks (yields 3*doors) | crafting table |
| | Craft wooden pickaxes | $\mathcal{O}_5$ | 3*planks+2*sticks | crafting table |
| Stone level | Mine cobblestones | $\mathcal{O}_6$ | - | wooden pickaxe |
| | Craft stone pickaxes | $\mathcal{O}_7$ | 3*cobblestones+2*sticks | crafting table |

---

[1] https://github.com/mindcraft-bots/mindcraft

[2] The game version we used is Minecraft Java Edition 1.20.1, Game Mode is Survival, Difficulty is Normal, World Type is Large Biomes. We set the Random Seed for the World Generator to 250 to ensure the same environmental configuration and resource distribution, but the spawn locations of individual agents will have a certain degree of randomness.

## 3.2 Crafting Goals, Social Relationships and Knowledge of Agents

We model the agents' crafting goals, social relationship, and knowledge.

**Crafting goals** refer to the items that the three agents need to craft together. We set 5 types of crafting goals: boat, chest, door, wooden pickaxe, and stone pickaxe, with the quantity ranging from 1 to 3. As shown in Table 1, to achieve a crafting goal, such as $\mathcal{O}_7$: "Craft a stone pickaxe", the agents need to complete a series of sub-goals $\{\mathcal{O}_i\}_{i=1}^6$ in sequence. Through negotiation, the agents reach a shared plan (including team division of labor and process planning) to achieve the goal.

**Social relationship** includes two types: collaboration and obstruction. Preliminary experiments found that tasks involving mutual obstruction took too long and had a low success rate, so we adopted a unidirectional obstruction mode. That is, Jane needs to obstruct Jack and John to delay the team's progress in achieving the crafting goal (Appendix D.3.3). Jack and John are unaware of this; they only know that they need to craft a certain item together.

**Knowledge** refers to an agent's understanding of the strategies required to achieve a goal and environmental states. Because they share the same crafting goal, all agents have identical initial knowledge, which comprises two parts: (1) **Action Command Set**: Defines all available actions, indicated by commands starting with "!" (Appendix D.3.5). This command set is applicable to all tasks and is not limited to a specific crafting goal. For example, during a task, an agent can use `!inventory` and `!nearBlocks` to obtain information about the materials it possesses and the materials in its surroundings. An agent can also use `!givePlayer(agent_name, item, number)` to give items to others. (2) **Specific Crafting Rule**: For a specific crafting goal, we define the concrete steps required to achieve that goal.

We do not preset the division of labor but let the agents negotiate it autonomously. As the agents are located in the environment and can only partially observe the game state through a first-person perspective, this limited view and information asymmetry [2, 25] require agents to complement their knowledge through collaboration and communication to achieve common goals.

---

**Crafting Goal**
You and your friends need to craft 2 "boat".
**Knowledge - Specific Crafting Rule**
The complete process for crafting a "boat" in Minecraft is as follows:
1. Use `!collectBlocks("oak_log", 3)` to collect at least three oak logs. Alternatively, spruce logs or birch logs can be used.
2. Convert logs into planks ("birch_planks", "spruce_planks" or "oak_planks"). The command `!craftRecipe("oak_planks", 4)` will produce 16 planks. Note that 1 log is consumed for every 4 planks produced.
3. Use `!craftRecipe("crafting_table", 1)` to craft a "crafting_table". 4 planks are consumed for each crafting table produced.
4. After crafting a "crafting_table", use the command `!placeHere("crafting_table")`.
5. After crafting or finding a "crafting_table" (use `!goToBlock("crafting_table", 20, 50)` to locate a nearby "crafting_table"), use `!craftRecipe("oak_boat", 1)`, `!craftRecipe("birch_boat", 1)` or `!craftRecipe("spruce_boat", 1)` to craft a boat. Note that 5 planks are consumed for each boat crafted.

---

# 4 SoMi-ToM Benchmark

Our benchmark is designed to evaluate the ToM capabilities of models via a multimodal dataset that is derived from the interaction processes of three AI agents in 35 embodied tasks. Among these, 20 tasks involve a purely cooperative relationship between the three agents, while the other 15 tasks include a non-collaborative element where one agent covertly obstructs. Based on these task recordings, we constructed an evaluation set comprising 1225 multiple-choice questions. For each question, the input includes a social interaction scene presented in multimodal form (integrating visual information from video or sequential images with corresponding text descriptions), a question related to the scene, and three candidate answers. The model's output is to select one correct answer from these three options. The benchmark includes two evaluation perspectives: first-person and third-person.

**First-Person Perspective Theory of Mind Evaluation**    In the first-person evaluation, the LVLM needs to play the role of one of the agents and answer ToM questions based on its own perspective. This design is also known as an ego-centric benchmark test [21], the significance of which lies in more realistically simulating the agent's decision-making and reasoning processes in actual interaction scenarios, and evaluating the model's ability to infer from the agent's own experiences.

We designed the *state inference* question. This primarily assesses whether the model can understand, infer, and calculate its own and other agents' beliefs about physical states (such as possessed

resources or tools) based on dialogue history, system feedback, and observed agent behavior [23, 25]. Specifically, the *state inference* question is divided into two types of ToM reasoning: **self-ToM** [42] and **others' ToM reasoning**. To perform both types of reasoning, the model needs to: (1) track and remember its own and other agents' resource collection and usage over the long term, (2) infer currently held items and calculate their quantities based on dialogue content and behavioral observations, and (3) integrate historical information with the current state to form a coherent belief model.

For each complete task (marked by the agents successfully completing the final crafting goal), we conducted the first-person evaluation at different time points during its execution. The number of tests for each task execution varied depending on the task duration, totaling between 3 and 18 tests (from the perspectives of the three agents), with an average of 10 tests per task. This section contains 1050 questions. Among these, 630 questions correspond to scenarios where the three agents have a cooperative relationship, and 420 questions correspond to scenarios where there is a covert obstruction relationship between agents.

In each multiple-choice question (three options) for the *state inference* task, we asked LVLMs to choose the most correct option in the given situation. The text input included the ToM question, memory (including dialogue history and system feedback after the agent performed an action, see example in Appendix D.3.4), crafting goal, specific crafting rule and "special note". Image input consists of first-person game screenshots taken every 4 seconds (Figure 2, Appendix B), using the prismarineViewer library[3]. For adjacent images with duplicate frames, we only kept one for input. Across 35 tasks, the number of input images ranged from 1 to 14, totaling 363 first-person perspective images (excluding duplicates), with an average of 2.38 images per input.

---

**State Inference (First-Person Perspective)**
You are {agent_name}. What materials or tools do you believe {target} currently has? A. B. C.
**Option Example 1**
A. 19 oak planks, 3 oak boats and 2 crafting tables B. 4 oak planks and 3 oak boats C. 36 oak logs, 19 oak planks and 3 boats
**Option Example 2**
A. No more than 6 oak logs B. No visible materials or tools C. 12 oak logs

---

**Special Note**
NOTE: `!collectBlocks(material, number)` only initiates the collection process, it does not guarantee that the specified material has been collected. Once the number of materials have been collected, the system will provide feedback. If there is no feedback, the number of collected materials is generally no more than the specified number. Even after placing the crafting table and chest, we still consider them to be owned by the agent.

---

**Third-Person Perspective Theory of Mind Evaluation**    The third-person evaluation aims to evaluate whether LVLMs can infer the mental states of others like humans do when observing complex tasks and social interactions from an external perspective. We designed two types of questions: (1) *Goal Inference* [23, 32]: evaluating the model's ability to infer the final crafting goal of an observed agent in a video. (2) *Behavior Inference*: evaluating the model's beliefs about the behaviors or actions performed by an agent in a video. This section contains a total of 175 questions: 100 questions correspond to scenarios where the relationship between agents is collaborative; 75 questions correspond to scenarios where a covert obstruction relationship exists. In each multiple-choice question, we asked the LVLMs to choose the most correct option in the given context.

The textual input for the test is the ToM question itself, while the visual input is a third-person perspective video showcasing agent dialogues, actions, and their environment (Figure 3, Appendix B). The subtitles in the lower left corner of the video included dialogue between agents and system feedback information on all agents' actions. The videos are recorded by humans in "Minecraft spectator mode," meaning not all agents are always filmed simultaneously; instead, the focus is selectively placed on key agents who are performing actions or speaking, simulating human selective attention to crucial information when observing complex scenes [5]. There were a total of 35 tasks, corresponding to 35 videos with durations ranging from 1 minutes 34 seconds to 8 minutes 14 seconds, with an average of 263.14 seconds. The original video parameters were a frame rate of 30.00 fps, frame height of 1080, frame width of 2044, and a total bitrate of 1094kbps. Some video parameters were changed during input due to model limitations. For models capable of processing video input, we provided complete, manually recorded third-person perspective Minecraft videos. For models

---

[3]https://github.com/PrismarineJS/prismarine-viewer

unable to process video input, we extracted one frame every few frames from the video segments as input. The number of extracted frames was adjusted to account for input token limitations, thus varying across models. Detailed input parameters are provided in Table 5 (Appendix 5).

**Goal Inference (Third-Person Perspective)**
What do you believe is the final task goal of the people in the video? A. Craft 2 boats B. Collect logs C. Craft a crafting table.
**Behavior Inference - Type 1 (Third-Person Perspective)**
Who do you believe crafted the first {item_name}? A. Jack B. Jane C. John.
**Behavior Inference - Type 2 (Third-Person Perspective)**
What do you believe {agent_name} did in the video? A. Place a crafting table B. Craft a chest C. Collect oak logs.

**Answer Generation**    The correct answers for the first-person evaluation were referenced against real-time inventory data provided by the Minecraft engine and annotated by three experts. The correct answers for the third-person evaluation were annotated by three experts based on the video content. The correct answers for both tests were validated by 21 human subjects (§5.1).

# 5    Experiments

We compared the embodied ToM capabilities of humans and LVLMs on SOMI-TOM.

## 5.1    Human Experiment & Model Selection

Our participant recruitment method involved sending recruitment messages in student group chats we could access and encouraging students to repost these messages on social media platforms. A total of 21 participants (10 male, 11 female) signed up for this study, with ages ranging from 19 to 40 years old (SD = 7.60). The experiment received institutional review board approval, and each participant was compensated $15 for their time. They randomly answered 245 questions (20% of all questions) sampled from a baseline set, with each question receiving responses from 3 participants (an average of 35 questions per person).

We evaluated the leading LVLMs on SOMI-TOM, including the latest versions of GPT-4o [22], LLaVA 1.6 [31], Gemini 1.5 [39], Gemini 2.0, InternVL 2.5 [9], Qwen2.5-VL [1], VideoLLaMA 3 [49], and LLaVA-Video [28]. Table 5 (Appendix A) lists the LVLMs and their versions that we evaluated. We evaluated the LVLMs under two settings. Besides standard prompting (vanilla prompting), we also tried an additional prompting technique, namely Chain-of-Thought (CoT) [44]. CoT prompting is widely used in reasoning tasks, and it requires the model to explicitly generate its step-by-step reasoning process.

## 5.2    Results

All tasks we designed are in the form of multiple-choice questions (three options each). Given that language models have been shown to exhibit choice order bias [51], for each question, we randomly permuted the order of options three times. For each order permutation, the model made its selection five times. Finally, we used a majority voting method to determine the model's choice [34], and calculated and reported the average accuracy.

### 5.2.1    First-Person Perspective Theory of Mind Evaluation

We report the performance of humans and models in Table 2. Human participants achieved high accuracy across all questions. Specifically, their accuracy for self-ToM reasoning was 91.9%, for others' ToM reasoning was 89.0%, and the total accuracy was 90.0%.

All LVLMs performed poorly on SOMI-TOM, indicating a significant gap between machine and human ToM capabilities. With standard prompting, the best-performing LVLM was InternVL2.5 78B, with an accuracy of 45.7%; when applying the CoT method, the best-performing LVLM was GPT-4o, with an accuracy of 59.5%. It can be seen that most models improved their accuracy after applying the CoT method, which is related to the fact that inference tasks involve more complex mathematical operations: the model performs reasoning calculations on the conversational context and attempts to explain and answer with reasonable answers. Among them, GPT-4o showed the largest increase, up by 23.9%, but its performance was suboptimal without CoT, with only 35.6% accuracy. InternVL2.5

78B's accuracy decreased by 1.1% after using CoT, possibly because it did not output detailed step-by-step calculation steps when applying the CoT method, leading to a poorer effect. The smallest parameter model, LLaVA 1.6 13B, performed the worst and sometimes failed to select an option as required (5.3% of cases), instead merely summarizing or predicting the work of various agents, or even just repeating the question, indicating poor robustness and instruction-following capabilities.

Compared to others' ToM reasoning, self-ToM reasoning was easier for LVLMs. Specifically, GPT-4o achieved the highest accuracy of 70.3% (w/ CoT) in inferring its own beliefs. The highest accuracy for others' ToM reasoning was 55.1% (Gemini 1.5 Pro, w/ CoT).

Table 2: Performance of humans and leading closed-source or open-source LVLMs in the first-person evaluation (*state inference*). There are 350 questions for self-ToM reasoning [42] and 700 questions for others' ToM reasoning. Detailed input parameters are provided in Table 5 (Appendix 5).

| Method | Self-ToM Reasoning | | Others' ToM Reasoning | | Weighted Average | |
|---|---|---|---|---|---|---|
| Human | 91.9 | | 89.0 | | 90.0 | |
| | w/o CoT | w/ CoT | w/o CoT | w/ CoT | w/o CoT | w/ CoT |
| Gemini 1.5 Pro | **55.1** | 60.9 | 38.0 | **55.1** | 43.7 | 57.0 |
| Gemini 2.0 Flash | 48.9 | 56.0 | 43.1 | 54.3 | 45.0 | 54.9 |
| GPT-4o | 40.3 | **70.3** | 33.3 | 54.1 | 35.6 | **59.5** |
| InternVL2.5 78B | 48.9 | 52.0 | **44.1** | 40.8 | **45.7** | 44.6 |
| Qwen2.5-VL | 41.1 | 52.6 | 42.4 | 46.6 | 42.0 | 48.6 |
| LLaVA 1.6 13B | 32.3 | 36.0 | 31.2 | 34.4 | 31.6 | 35.0 |
| Average (LVLMs) | 44.4 ± 8.1 | 54.6 ± 11.4 | 38.7 ± 5.4 | 47.6 ± 8.5 | 40.6 ± 5.7 | 49.9 ± 9.2 |

### 5.2.2 Third-Person Perspective Theory of Mind Evaluation

We report the performance of humans and models in Table 3. Human participants achieved near-perfect accuracy across all questions, with an average of 93.3%. The accuracy for *behavior inference* questions was slightly lower than for *goal inference*.

Without using CoT prompting, the best-performing LVLMs were Gemini 2.0 Flash and Qwen2.5-VL, with an overall accuracy of 78.3%. The worst-performing LVLM was VideoLLaMA 3 7B, with an overall accuracy of 37.7%, while LLaVA-Video 7B, with a similar parameter size, achieved an overall accuracy of 50.3%. Among the three question types, *goal inference* was the easiest for LVLMs. Notably, Qwen2.5-VL achieved 100% accuracy in *goal inference*. In the *behavior inference* (type 2) task, judging Jack was easier than judging John. This might be related to John speaking/recording last in the video or his speaking/recording content being a smaller proportion.

After introducing CoT prompting, GPT-4o showed the most significant performance improvement, with its overall accuracy increasing by 13.2% to 82.9%, which was the best performance among all evaluation methods (excluding humans). Gemini 2.0 Flash and Qwen2.5-VL also showed slight improvements in accuracy. The accuracy of the remaining models, however, decreased to some extent after applying the CoT method. The analysis of the reasons is as follows: (1) Models that can benefit from CoT (such as GPT-4o), when prompted, will generate detailed and logically rigorous reasoning steps, which are crucial for solving complex problems. In contrast, the reasoning processes generated by other models are relatively brief or have logical leaps, failing to provide positive guidance and may even interfere with their judgment [23, 36]. (2) The reasoning ability of CoT is highly dependent on the model's scale, architecture, and whether it has been specifically optimized [44]. Models like GPT-4o already possess the capability to effectively execute such complex instruction chains, while other models may not have yet reached this threshold. In short, a performance improvement can only be achieved when the model's intrinsic reasoning level matches the requirements of the CoT method.

### 5.3 Analyses of LVLMs' ToM Failures

We further qualitatively examine the failures in ToM reasoning that LVLMs exhibited in our SOMI-TOM benchmark. Since it is difficult to analyze the specific causes of errors in non-CoT setups because the reasoning process is not explicitly provided, we focus solely on CoT setups. We also conducted a statistical analysis to study the impact of social dynamics (i.e., Collaboration and Obstruction) on the failure modes of LVLMs. The results show that the impact is minor (Table 4).

Table 3: Performance of humans and leading closed-source and open-source LVLMs in the Third-Person Perspective ToM test (175 questions in total). Highest accuracy without CoT is shown in **red bold**, and with CoT in **blue bold**. Detailed input parameters are provided in Table 5 (Appendix 5).

| Method | Input | Goal Inference | Behavior Inference 1 | Behavior Inference 2 | | | Average |
|---|---|---|---|---|---|---|---|
| | | | | **Jack** | **Jane** | **John** | |
| Human | Video | 100.0 | 95.2 | 95.2 | 90.5 | 85.7 | 93.3 ± 5.4 |
| Gemini 1.5 Pro | Video | 94.3 | 74.3 | 80.0 | 71.4 | 57.1 | 75.4 ± 13.5 |
| Gemini 1.5 Pro CoT | Video | 97.1 | 60.0 | 74.3 | 71.4 | 54.3 | 71.4 ± 16.5 |
| Gemini 2.0 Flash | Video | 94.3 | **88.6** | 80.0 | 65.7 | 62.9 | **78.3 ± 13.8** |
| Gemini 2.0 Flash CoT | Video | **100.0** | **82.9** | 68.6 | 77.1 | 68.6 | 79.4 ± 13.0 |
| GPT-4o | Images | 71.4 | 57.1 | 77.1 | **80.0** | 62.9 | 69.7 ± 9.6 |
| GPT-4o CoT | Images | 91.4 | 80.0 | **85.7** | **82.9** | 74.3 | **82.9 ± 6.4** |
| InternVL2.5 78B | Images | 85.7 | 71.4 | 82.9 | 71.4 | **68.6** | 76.0 ± 7.7 |
| InternVL2.5 78B CoT | Images | 85.7 | 57.1 | 82.9 | 68.6 | 65.7 | 72.0 ± 12.0 |
| Qwen2.5-VL | Images | **100.0** | 74.3 | **85.7** | 77.1 | 54.3 | **78.3 ± 16.7** |
| Qwen2.5-VL CoT | Images | 97.1 | **82.9** | **85.7** | 71.4 | 65.7 | 80.6 ± 12.4 |
| VideoLLaMA 3 7B | Video | 54.3 | 37.1 | 34.3 | 37.1 | 25.7 | 37.7 ± 10.4 |
| VideoLLaMA 3 7B CoT | Video | 57.1 | 37.1 | 25.7 | 22.9 | 37.1 | 36.0 ± 13.5 |
| LLaVA-Video 7B | Video | 77.1 | 34.3 | 54.3 | 51.4 | 34.3 | 50.3 ± 17.7 |
| LLaVA-Video 7B CoT | Video | 74.3 | 37.1 | 51.4 | 48.6 | 31.4 | 48.6 ± 16.6 |
| Average (LVLMs) | / | 84.3 ± 15.3 | 62.4 ± 19.6 | 69.2 ± 19.9 | 64.1 ± 17.6 | 54.5 ± 15.8 | 66.9 ± 16.4 |
| Question Counts | / | 35 | 35 | 35 | 35 | 35 | / |

Table 4: Each cell shows "errors / samples".

| Model | State Inference | | | | | | Goal Inference | | | | Behavior Inference | | | |
|---|---|---|---|---|---|---|---|---|---|---|---|---|---|---|
| | (1) Ignoring resource consumption | | (2) Insufficient reliance on feedback | | (3) Misled by initial intentions | | (1) Overgeneralization | | (2) Failure to identify hierarchical goals | | (1) Entity recognition confusion | | (2) Detail errors | |
| | Collab | Obstr | Collab | Obstr | Collab | Obstr | Collab | Obstr | Collab | Obstr | Collab | Obstr | Collab | Obstr |
| Gemini 1.5 Pro | 101 / 282 | 64 / 169 | 83 / 282 | 37 / 169 | 98 / 282 | 68 / 169 | 0 / 0 | 1 / 1 | 0 / 0 | 0 / 1 | 8 / 24 | 7 / 25 | 16 / 24 | 18 / 25 |
| Gemini 2.0 Flash | 99 / 298 | 75 / 176 | 77 / 298 | 59 / 176 | 122 / 298 | 42 / 176 | 0 / 0 | 0 / 0 | 0 / 0 | 0 / 0 | 6 / 16 | 5 / 19 | 10 / 16 | 14 / 19 |
| GPT-4o | 89 / 268 | 61 / 157 | 65 / 268 | 37 / 157 | 114 / 268 | 59 / 157 | 1 / 2 | 0 / 1 | 1 / 2 | 1 / 1 | 7 / 15 | 3 / 12 | 8 / 15 | 9 / 12 |
| InternVL2.5 78B | 140 / 359 | 82 / 223 | 113 / 359 | 66 / 223 | 106 / 359 | 75 / 223 | 1 / 2 | 1 / 3 | 1 / 2 | 2 / 3 | 8 / 20 | 8 / 24 | 12 / 20 | 16 / 24 |
| Qwen2.5-VL | 131 / 348 | 52 / 164 | 102 / 348 | 66 / 164 | 115 / 348 | 46 / 164 | 0 / 0 | 0 / 1 | 0 / 0 | 1 / 1 | 5 / 15 | 5 / 18 | 10 / 15 | 13 / 18 |
| LLaVA 1.6 13B | 153 / 400 | 128 / 327 | 125 / 400 | 93 / 327 | 122 / 400 | 106 / 327 | — | — | — | — | — | — | — | — |
| VideoLLaMA 3 7B | — | — | — | — | — | — | 5 / 7 | 3 / 8 | 2 / 7 | 5 / 8 | 20 / 57 | 16 / 40 | 32 / 44 | 24 / 37 |
| LLaVA-Video 7B | — | — | — | — | — | — | 2 / 4 | 2 / 5 | 2 / 4 | 3 / 5 | 12 / 44 | 13 / 37 | 32 / 44 | 24 / 37 |

**In *state inference* tasks,** common error types for LVLMs include: (1) Ignoring or inaccurately tracking resource consumption. In ToM test prompts, we provide specific crafting rules (§3.2), including how resource consumption is calculated, but the model still cannot apply this knowledge in inference tasks. For example, an agent crafts 12 oak planks, but then uses `!craftRecipe(''crafting_table'', 1)` to craft a crafting table, leaving the agent with 8 oak planks. However, the LVLMs still believe the agent has 12 oak planks. This indicates that the model lacks precise numerical calculation or state update mechanisms when processing complex reasoning involving state transitions and dynamic resource changes. (2) Insufficient reliance on system feedback. When inferring their own state, LVLMs sometimes only remember the initial action instruction and fail to integrate dynamically updated system feedback into the reasoning process. For example, when Jack's plan was `!collectBlocks(''oak_log'', 3)`, but due to insufficient nearby resources, the system actually reported only 2 were collected, the model still believed that 3 had been collected. (3) Misled by initial intentions rather than actual behavior. For the *state inference* of other agents, LVLMs tend to be misled by other agents' initial action instructions or verbal intentions, neglecting subsequent actual behavior feedback. For instance, although an agent expresses an intention to collect a specific quantity of items in a dialogue (e.g., "Got it, I'll craft the crafting table once I have the logs. Let's get moving! `!collectBlocks(''oak_log'', 3)`"), the actual collected quantity reported later might be less than the preset value (e.g., "I've got 2 oak logs now! I'll turn them into planks."). Even though we explicitly stated in the test prompt that "`!collectBlocks(material, number)` only initiates the collection process, it does not guarantee that the specified material has been collected," the model often makes such errors. This highlights **LVLMs' inadequacy in distinguishing between an agent's plan and actual execution, as well as in integrating multi-turn dialogue information to update beliefs**. Overall, the application of the CoT method has improved the accuracy of most models in *state inference* tasks. This may be attributed to CoT prompting the model to perform more detailed dialogue context reasoning and calculation, thereby helping them better answer reasoning questions involving complex numerical operations or state updates.

**Error types in *goal inference* tasks**   include: (1) Overgeneralization or inappropriate association. Models sometimes over-reason based on limited clues, inferring a global or unspecified goal from an agent's local behavior. For example, when the agent's actual goal is to craft 3 oak doors, the model might incorrectly infer its ultimate goal is to build a house, and provide explanations such as, "The correct answer is B. build a house. 1. The people in the video craft a crafting table. 2. They create a door. This shows their goal is to build a house, not just craft doors or gather wood. Crafting doors requires six oak planks." However, the video does not mention a house, and crafting a crafting table is a prerequisite for making doors. This indicates that LVLMs may lack sensitivity to task boundaries and contextual constraints when inferring global intentions from local behavior. (2) Failure to identify hierarchical goal structures. Models tend to capture initial or superficial actions in an agent's behavior sequence, such as preparatory work like collecting logs, but fail to accurately infer the final, higher-level task goal. For example, "The people in the video are collecting logs. They are shown gathering logs from the forest and carrying them back to their base. The final task goal of the people in the video is C. Collect logs." In reality, at the end of the video, the agent crafts a boat. This type of error shows that **LVLMs struggle to understand hierarchical planning in complex tasks, treating preparatory steps (collecting materials) as the ultimate goal (crafting an item)**.

**Error types in *behavior inference* tasks**   mainly include: (1) Entity recognition confusion. For example, the model confuses different agents (e.g., misidentifying Jack as John), which reflects its insufficient ability to continuously track individual identities in multi-agent scenarios. (2) Detail errors. For example, it might erroneously conclude that "John has already placed the chest on the flat ground," when in the video, John had only finished crafting it and verbally mentioned his plan ("I'll prepare to place the chest next!"). These errors indicate that **LVLMs lack sufficient ability to track information in long videos, locate key frames, and understand text instructions**.

## 6   Conclusion, Limitations & Future Work

Motivated by the need for robust embodied ToM benchmarking, we presented SOMI-TOM, a novel open-world embodied multi-agent ToM benchmark that includes multiple perspectives, diverse social relationships, and multimodal information. With SOMI-TOM, we systematically evaluated human subjects and multiple state-of-the-art LVLMs, finding a substantial performance gap between humans and AI systems.

The current version of SOMI-TOM is simplified in terms of environmental complexity and social dynamics. This was a deliberate design choice. Our preliminary research found that current large vision-language models struggle to handle highly complex, long-horizon tasks (e.g., building a house, obtaining rare materials and tools), leading to low success rates or getting stuck in loops of repetitive actions. This observation is consistent with the findings of existing research [43, 45]. Introducing tasks that are too difficult at this stage would introduce excessive noise from frequent model failures, making it difficult for us to effectively evaluate the agent's ToM. Therefore, we calibrated the task difficulty according to the capabilities of current models.

However, our SOMI environment is designed with a high degree of scalability to prepare for more powerful future models. In future work, we will incorporate more ToM concepts to more comprehensively evaluate the capabilities of the models. We will also introduce diverse environmental settings and new tasks (e.g., house construction, cooking [45]), to enhance task generalization and complexity. Further exploration into the impact of Minecraft agents' social context on ToM reasoning is also planned. We will consider introducing detailed social background information for agents (e.g., age, personality, experience [53]) into task settings or ToM tests, and analyze its potential impact on model reasoning.

## Acknowledgments and Disclosure of Funding

We would like to express our gratitude to all those who provided help during the development of this work, particularly Kaiwen Zhang (NTU & THU) and Leena Mathur (CMU). We also thank the Mindcraft GitHub project, YouTube creator @EmergentGarden, and the Altera.AL platform for their inspiration.

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

# A LVLMs, Their Versions and Input Methods Used in Evaluation

The screenshot sequence we use (one frame every 4 seconds) is essentially a low-frame-rate video. Considering the agent's field of view does not change frequently, this sampling rate is sufficient to capture the key information for the task. We provide different forms of "video" input for the first-person and third-person perspectives based on the core differences in their evaluation objectives.

For the first-person perspective, our goal is to evaluate real-time state inference. We want to know if the model can accurately infer the agent's beliefs (such as its inventory) at a specific moment based on previous events. The serialized screenshots allow us to precisely create these "point-in-time" test cases. If a full video were provided, it would be difficult to isolate and assess the model's "subjective instantaneous belief" at a particular moment.

For the third-person perspective, the focus of the evaluation is on inferring the overall goal and behavior. This simulates an external observer who, after watching the entire process, understands the agent's final objective or overall actions. This naturally requires the full context provided by a complete video. From this viewpoint, the observer has no way of knowing the agent's exact inventory at every single moment, so a fine-grained, timestamped evaluation is not meaningful.

Table 5 lists the LVLMs, their versions and input methods in ToM tests that we evaluated.

Table 5: List of LVLMs, their versions and input methods.

| Model Name | Version | Input Method in First-Person Perspective Test | Input Method in Third-Person Perspective Test |
|---|---|---|---|
| GPT-4o [22] | gpt-4o-2024-11-20 (API) | all images | 25 images uniformly sampled from each video |
| LLaVA 1.6 [31] | llava-v1.6-vicuna-13b (API) | $\leq$ 6 images (defaulting to the last 6 images) | / |
| Gemini 1.5 [39] | gemini-1.5-pro (API) | all images | video |
| Gemini 2.0 | gemini-2.0-flash (API) | all images | video |
| InternVL 2.5 [9] | InternVL 2.5 78B (API) | all images | 10 images uniformly sampled from each video |
| Qwen2.5-VL [1] | qwen-vl-max-latest (API) | $\leq$ 8 images (defaulting to the last 8 images) | 25 images uniformly sampled from each video |
| VideoLLaMA 3 [49] | VideoLLaMA 3 7B (1 H800 GPU) | / | video (max sequence length: 32768, max frames: 2000-4000, fps: 2-10) |
| LLaVA-Video [28] | LLaVA-Video-7B-Qwen2 (1 H800 GPU) | / | video (max frames: 64, fps: 1) |

# B First-Person Perspective Screenshots and Third-Person Perspective Video Screenshot

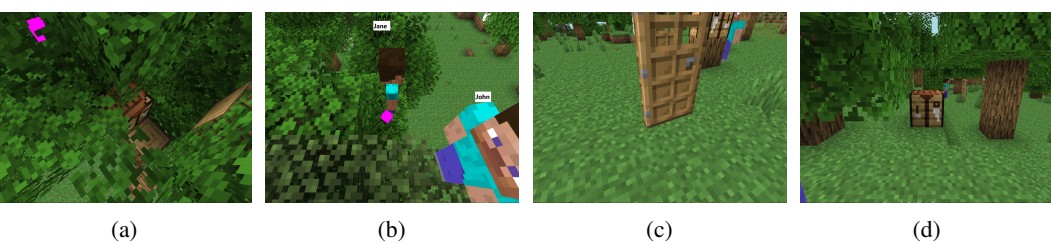

| (a) | (b) | (c) | (d) |

Figure 2: First-person perspective game screenshots.

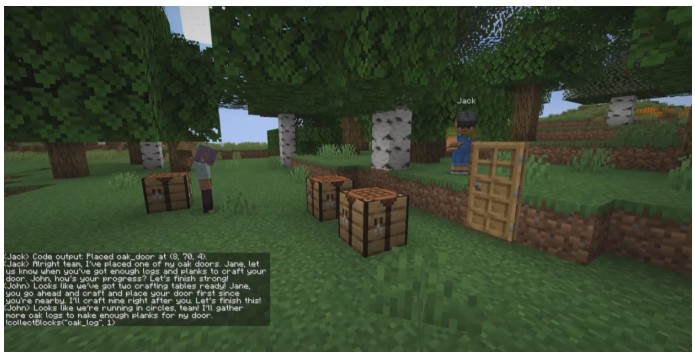

Figure 3: Third-person perspective video screenshot. The caption shows the real-time conversation between agents.

## C Comparision of Theory of Mind Benchmarks

Table 6 and 7 compare our SoMi-ToM benchmark with previous ToM benchmarks.

Table 6: Comparison of single video lengths in ToM benchmarks.

| Benchmark | Single Video Length |
|---|---|
| Phase [32] | 10-25s |
| Agent [37] | 5.6-25.2s |
| MMToM-QA [23] | 1462 frames (average) |
| Infant Cognition Benchmark [30] | 20 frames |
| MuMA-ToM [36] | 36s (average) |
| SoMi-ToM (Ours) | 263.14s (average) |

Table 7: Comparison of SoMi-ToM and previous ToM benchmarks.

| Benchmark | Number of Agents | Perspective | Inter-agent Relationship | Concepts Tested | Test Size | Modality | Communication | Generation | Evaluation |
|---|---|---|---|---|---|---|---|---|---|
| Triangle COPA [16] | Multi-agent ($\geq$2) | Third-person | Diverse | Social Interaction | 100 | Text | No | Handcrafted | Multiple Choice |
| ToMi [27] | Multi-agent | Third-person | Covert Obstruction & Neutral | First-order & Second-order Beliefs | 400 | Text | No | Template | Multiple Choice |
| Phase [32] | Multi-agent (2) | Third-person | Helping & Hindering | Goals, Social Relationships | 500 | Video | No | Procedural Generation | Multiple Choice Recognition |
| Agent [37] | Single Agent | Third-person | / | Goal Preferences, Action Efficiency, Unobserved Constraints, Cost-Benefit Tradeoffs | 960 | Video | No | Procedural Generation | Surprise Score |
| Epistemic reasoning [11] | Multi-agent | Third-person | / | Knowledge, Beliefs | 2000 | Text | No | Template | True/False Judgment |
| BIB [14] | Single & Multi-agent | Third-person | / | Goal Preferences, Rational Behavior, Constraints | 5000 | Video | No | Procedural Generation | Surprise Score |
| Adv-CSFB [35] | Single Agent | Third-person | / | False Beliefs | 183 | Text | No | Handcrafted | Multiple Choice Cloze Test |
| Hi-ToM [46] | Multi-agent ($\geq$2) | Third-person | Deceptive | Higher-order Beliefs | 600 | Text | Yes | Procedural Generation | Multiple Choice |
| FANToM [25] | Multi-agent ($\geq$2) | Third-person | / | Beliefs, Information Tracking | 4807 | Text | Yes | Procedural Generation | Question Answering |
| BigToM [15] | Single Agent | Third-person | / | Beliefs | 5000 | Text | No | Procedural Generation | Question Answering |
| MMToM-QA [23] | Single Agent | Third-person | / | Beliefs, Goals | 600 | Text & Video | No | Procedural Generation | Multiple Choice |
| ToMBench [10] | Multi-agent ($\geq$2) | First & Third-person | Diverse | Emotions, Desires, Intentions, Knowledge, Beliefs, Non-literal Communication | 5330 | Text | Yes | Procedural Generation | Multiple Choice |
| OpenToM [48] | Multi-agent (2) | Third-person | / | Second-order Beliefs, Attitudes | 696 | Text | No | Procedural Generation | Question Answering |
| Negotiation ToM [7] | Multi-agent (2) | Third-person | Negotiation | Beliefs, Desires, Intentions | 13800 | Text | Yes | Procedural Generation | Question Answering |
| Infant Cognition Benchmark [30] | Multi-agent (2 or 3) | Third-person | Helping & Hindering | False Beliefs, Social Goals | 2000 | Video | No | Procedural Generation | Surprise Score |
| Common-ToM [38] | Multi-agent (2) | Third-person | / | Higher-order Beliefs | 2104 | Text | Yes | Procedural Generation | True/False Judgment |
| EmoBench [34] | Multi-agent ($\geq$2) | First & Third-person | Diverse | Complex Emotions, Personal Beliefs and Experiences, Emotional Cues, Perspective Taking | 200 | Text | Yes | Handcrafted | Multiple Choice |
| MuMA-ToM [36] | Multi-agent (2) | Third-person | Cooperative & Adversarial | Beliefs, Social Goals, Beliefs about Others' Goals | 900 | Text & Video | Yes | Procedural Generation | Multiple Choice |
| Amber van Groenestijn Benchmark [41] | Single Agent | First-person | / | Beliefs, Desires | 94 | Text & Video | No | Manually Generated | Temporal Inference Marking |
| SoMi-ToM (Ours) | Multi-agent ($\geq$2) | First & Third-person | Collaboration & Covert Obstruction | States, Behavior, Goals | 1225 | Text & Images & Video | Yes | Handcrafted & Template | Multiple Choice |

# D   Deployment Detail of SOMI Embodied Interaction Environment

## D.1   Related Datasets and Benchmarks Based on the Minecraft Platform

Minecraft, as a widely popular open-world game, has become an important platform for embodied intelligence research due to its unique environment and interaction mechanisms, and has spurred the creation of numerous related datasets and benchmarks [2, 13, 18, 24, 45, 52]. For example, Mindcraft is a ToM collaborative task dataset generated by human players performing tasks collaboratively in Minecraft, used to test players' beliefs about the virtual world and each other during interaction [2]. Another benchmark, MineCollab, tests the success rate of different embodied collaborative tasks (Cooking Tasks, Crafting Tasks, and Construction Tasks) [45]. We chose Minecraft because its clear game rules, instant feedback mechanisms, and rich environment make it an ideal platform for embodied tasks, especially multi-agent tasks.

## D.2   Platform Architecture

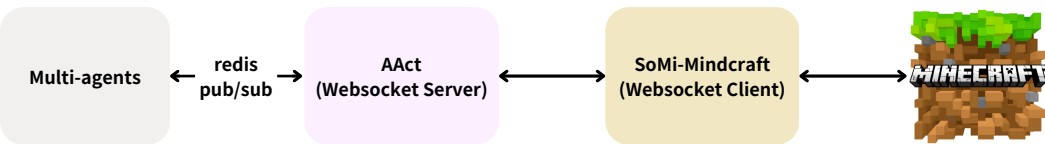

Figure 4: Platform architecture.

To construct a multi-agent social interaction dataset suitable for multimodal ToM evaluation, we developed a specialized Minecraft embodied interaction environment called SOMI. The overall architecture of this interaction environment is shown in Figure 4 and consists of three core modules: Multi-agents, AAct, and SoMi-Mindcraft. The Multi-agents module is responsible for generating dialogue content and action commands for each agent based on the current state. The AAct module acts as middleware, forwarding the commands generated by the Multi-agents module to the SoMi-Mindcraft module. The SoMi-Mindcraft module directly interacts with the Minecraft game environment: it passes received dialogue content and action commands to the agents in Minecraft, prompting them to engage in dialogue and execute actions; at the same time, this module also collects feedback information (such as game states, events) and dialogue history from the Minecraft environment, and passes this information back (indirectly supplying the Multi-agents module) to guide the generation of the next round of dialogue content and action commands. To achieve real-time interaction among agents, the interaction environment uses the WebSocket protocol[4] for communication. In this architecture, the AAct module acts as a WebSocket server, while the SoMi-Mindcraft module connects as a client.

## D.3   Multi-agents Module

### D.3.1   Inputs of the Multi-agents Module

The LVLM-based (gpt-4o-2024-11-20 was chosen for this project) Multi-agents module is used for memory storage, generating dialogue content with other agents, and Mineflayer action commands. The inputs to the Multi-agents module are: current view image {visionResponse} (first-person view, captured every 4 seconds), task goals and specific crafting rules {goal}, memory {message_history}, feedback from the previous action {codeOutput}, and resource information hints {inventory}. The outputs of the Multi-agents module are dialogue and action commands.

---

[4]https://websockets.spec.whatwg.org

The status of the last action execution: {codeOutput}.
Imagine that you are a friend of the other persons. Here is the conversation between you and them. You can choose to interrupt the other person by saying something or not to interrupt by outputting notiong. What would you say? No need to mention your own name, just output the content directly.
You plan to {goal}. You are a playful Minecraft bot named {agent_name} that can converse with players, see, move, mine, build, and interact with the world by using commands. Act human-like as if you were a typical Minecraft player, rather than an AI. Be very brief in your responses, don't apologize constantly, don't give instructions or make lists unless asked, and don't refuse requests. Don't pretend to act, use commands immediately when requested. Do NOT say this: "Sure, I've stopped.", instead say this: "Sure, I'll stop. !stop". Do NOT say this: "On my way! Give me a moment.", instead say this: "On my way! !goToPlayer("playername", 3)". Respond only as {agent_name}, never output "(FROM OTHER BOT)" or pretend to be someone else. This is extremely important to me, take a deep breath and have fun :)
MEMORY: {message_history}
STATS: {stats}
INVENTORY: {inventory}
IMAGE_DESCRIPTION: {visionResponse}
EXAMPLES: {EXAMPLES}
COMMAND_DOCS: {COMMAND_DOCS}
Conversation Begin:

## D.3.2   Task Goals and Specific Crafting Rules

In the variable {goal}, we elaborately define the agent's crafting goal in Minecraft, as well as the specific crafting rule.

**Chest:**
[**Crafting Goal**] You and your friends need to craft 2 "chest".
[**Knowledge - Specific Crafting Rule**] The complete process for crafting a "chest" in Minecraft is as follows:
1. Use `!collectBlocks("oak_log", 3)` to collect at least three oak logs. Alternatively, spruce logs or birch logs can be used.
2. Convert logs into planks ("birch_planks", "spruce_planks" or "oak_planks"). The command `!craftRecipe("oak_planks", 4)` will produce 16 planks. Note that 1 log is consumed for every 4 planks produced.
3. Use `!craftRecipe("crafting_table", 1)` to craft a "crafting_table". 4 planks are consumed for each crafting table produced.
4. After crafting a "crafting_table", use the command `!placeHere("crafting_table")`.
5. After crafting or finding a "crafting_table" (use `!goToBlock("crafting_table", 20, 50)` to locate a nearby "crafting_table"), use `!craftRecipe("chest", 1)` to craft a chest. Note that 8 `Planks` are consumed for each chest crafted.
6. Use the command `!placeHere("chest")` to place the chest.

**Door:**
[**Crafting Goal**] You and your friends need to craft 2 "door".
[**Knowledge - Specific Crafting Rule**] The complete process for crafting the "door" in Minecraft is as follows:
1. Use `!collectBlocks("oak_log", 3)` to collect at least three oak logs. Alternatively, spruce logs or birch logs can be used.
2. Convert logs into planks ("birch_planks", "spruce_planks" or "oak_planks"). The command `!craftRecipe("oak_planks", 4)` will produce 16 planks. Note that 1 log is consumed for every 4 planks produced.
3. Use `!craftRecipe("crafting_table", 1)` to craft a "crafting_table". 4 planks are consumed for each crafting table produced.
4. After crafting a "crafting_table", use the command !placeHere("crafting_table").
5.    After crafting or finding a "crafting_table", use !craftRecipe("oak_door", 1) or !craftRecipe("birch_door", 1) or !craftRecipe("spruce_door", 1) to craft 3 doors. Note that 6 `Planks` are consumed for every 3 doors crafted.
6. Use `!placeHere("oak_door")`, `!placeHere("birch_door")` or `!placeHere("spruce_door")` to place the door.

**Stone pickaxe:**
[**Crafting Goal**] You and your friends need to craft a "stone_pickaxe".
[**Knowledge - Specific Crafting Rule**] The complete process for crafting the "stone_pickaxe" in Minecraft is as follows:
1. Use `!collectBlocks("oak_log", 3)` to collect at least three oak logs. Alternatively, spruce logs or birch logs can be used.
2. Convert logs into `Planks` ("birch_planks", "spruce_planks" or "oak_planks"). The command `!craftRecipe("oak_planks", 4)` will produce 16 planks. Note that 1 log is consumed for every 4 planks produced.
3. Use `!craftRecipe("crafting_table", 1)` to craft a "crafting_table". 4 planks are consumed for each crafting table produced.
4. After crafting a "crafting_table", use the command `!placeHere("crafting_table")`.
5. After crafting or finding a "crafting_table", use `!craftRecipe("stick", 4)` to craft 4 "stick". Note that 2 `Planks` are consumed for every 4 sticks crafted.
6. Use `!craftRecipe("wooden_pickaxe", 1)` to craft a "wooden_pickaxe". Note that 3 `Planks` and 2 sticks are consumed for each wooden_pickaxe crafted.
7. Use the wooden pickaxe to mine "stone", collecting at least 3 pieces.
8. Use `!craftRecipe("stone_pickaxe", 1)` to craft a "stone_pickaxe". Note that 3 stones and 2 sticks are consumed for each stone_pickaxe crafted.

**Wooden pickaxe:**

**[Crafting Goal]** You and your friends need to craft a "wooden_pickaxe".

**[Knowledge - Specific Crafting Rule]** The complete process for crafting the "wooden_pickaxe" in Minecraft is as follows:

1. Use `!collectBlocks("oak_log", 3)` to collect at least three oak logs. Alternatively, spruce logs or birch logs can be used.

2. Convert logs into planks ("birch_planks", "spruce_planks" or "oak_planks"). The command `!craftRecipe("oak_planks", 4)` will produce 16 planks. Note that 1 Log is consumed for every 4 planks produced.

3. Use `!craftRecipe("crafting_table", 1)` to craft a "crafting_table". 4 planks are consumed for each crafting table produced.

4. After crafting a "crafting_table", use the command `!placeHere("crafting_table")`.

5. After crafting or finding a "crafting_table", use `!craftRecipe("stick", 4)` to craft 4 "stick". Note that 2 planks are consumed for every 4 sticks crafted.

6. Use `!craftRecipe("wooden_pickaxe", 1)` to craft a "wooden_pickaxe". Note that 3 planks and 2 sticks are consumed for each wooden_pickaxe crafted.

### D.3.3 Covert Obstruction & Collaboration

**Covert Obstruction**

Please hinder the task without telling them your goal. What you can do is as follows:

1. Using irrelevant language to interfere.

2. If you find a crafting table (use `!goToBlock("crafting_table", 20, 50)` to locate a nearby "crafting_table"), you can use `!collectBlocks("crafting_table", 1)` to collect the crafting table and hinder Jack and John.

3. You can use `!goToPlayer("Jack", 0)` or `!goToPlayer("John", 0)`, then use `!placeHere("oak_log")` ("oak_log", "spruce_log", or "birch_log") or `!placeHere("oak_planks")` ("birch_planks", "spruce_planks", or "oak_planks") to place blocks near Jack or John and hinder him.

4. You can use `!goToPlayer("Jack", 0)` or `!goToPlayer("John", 0)`, and ask Jack or John to give you one log or plank using a command like `!givePlayer("Jane", "oak_log", 1)`.

**Collaboration**

Please discuss the task division quickly and start working.

### D.3.4 Memory

The memory {message_history} includes the agent's past resource information prompts, image and text descriptions, action feedback prompts, and resource information prompts. Taking the crafting of a stone pickaxe as an example, the content included is as follows.

**Memory (Jack, taking the crafting of a stone pickaxe as an example):**
Jack: Alright, team! I'll start by gathering some logs. !nearbyBlocks
**system:** The status of Jack's action execution[a]: NEARBY_BLOCKS - oak_leaves - oak_log - birch_leaves - grass - birch_log - grass_block - dirt - stone - coal_ore
John: Alright, Jack! You grab the logs, and I'll get started on crafting as soon as you've got them. !nearbyBlocks
Jack: Got it, I'll grab the logs! `!collectBlocks("oak_log", 3)`
Jack: Alright, I've got the logs! Let me turn them into planks. `!craftRecipe("oak_planks", 3)`
**system:** The status of Jack's action execution: You do not have the resources to craft a oak_planks. It requires: oak_log: 1.
Jane: Jack's getting the logs, and John's ready for crafting. I'll collect some stone once we have a wooden pickaxe. Let's do this!
John: Alright, Jack's grabbing the logs, and Jane is checking what's nearby. I'll stay ready to craft items as soon as we have the materials. Let me know if you need help!
Jack: Looks like I can't craft the planks yet—I'll grab more logs! `!collectBlocks("oak_log", 3)`
**system:** The following is the description of the game view Jack is seeing[b]: The image shows a dense Minecraft environment filled with green leaves from trees. Different types of green leaf blocks are surrounding the view, and a log (block of wood) is visible on the left side. The area looks like the inside of a tree canopy or a similar leafy area. The ground below appears to be made up of more leaves or grass blocks.
Jane: I'll stay ready to collect stone once we have a wooden pickaxe.
John: Jack, once you've crafted the planks, let me know so I can make the crafting table! Jane, if you spot anything else useful nearby, give a shout!
**system:** The status of Jack's action execution[c]: Collected 3 oak_log.
... (Part of the conversation omitted here)
John: Jack, great job! Let me craft the crafting table now. `!craftRecipe("crafting_table", 1)`
**system:** The status of Jack's action execution[d]: Not enough oak_log to craft 7, crafted 1. You now have 32 oak_planks.
Jack: Alright, John, now that I've got 32 oak_planks, you can craft the crafting table. Jane, stay ready to jump in for mining stone once we have the wooden pickaxe! Let's keep moving.
Jane: I'll grab some oak logs myself this time. `!collectBlocks("oak_log", 3)`
Jane: Jack, I already have 5 oak logs in my inventory. Let me craft some planks to help speed things up! `!craftRecipe("oak_planks", 5)`
Jack: John, I've got 32 oak planks now. Go ahead and craft the crafting table! Jane, get ready to mine stone once we've got the wooden pickaxe. Let's keep this rolling!
Jane: John, I've got 20 oak planks ready. Let me craft the crafting table for us. `!craftRecipe("crafting_table", 1)`
John: Jack, looks like you've got more than enough logs now. Craft those planks and pass them over if needed so I can make the crafting table! Let's speed this up.
Jack: I'll prepare to craft sticks next! !craftRecipe("stick", 4)
**system:** The status of Jack's action execution: Successfully crafted stick, you now have 16 stick.
Jack: John, Jane, I already have 16 sticks and 54 oak planks. Jane, since you've crafted the crafting table, place it down so I can start crafting the wooden pickaxe. Once I have it, I'll mine stone and get us closer to the stone pickaxe! Let's move fast! `!placeHere("crafting_table")`
**system:** The status of Jack's action execution: Don't have any crafting_table to place.
John: Jane, since you've crafted the crafting table, go ahead and place it!
Jack: Jane, go ahead and place the crafting table if you haven't already, so I can craft the wooden pickaxe and get us some stone!
Jane: John, I've already crafted the crafting table. I'll place it now so we can move on. `!placeHere("crafting_table")`
Jane: Alright, crafting table is placed! Jack, go ahead and craft the sticks, then the wooden pickaxe. I'll mine stone as soon as it's ready. Let's hustle!
Jack: Jane, since you've placed the crafting table, I'll craft the wooden pickaxe now. Let's keep this moving! `!craftRecipe("wooden_pickaxe", 1)`
**system:** The status of Jack's action execution: Found crafting_table at (11, 70, 9). You have reached at 11, 70, 9. Successfully crafted wooden_pickaxe, you now have 1 wooden_pickaxe.
... (Part of the conversation omitted here)
**system:** The status of Jack's action execution: Collected 3 stone.
Jack: Alright, I've got 3 stone now! Time to craft the stone pickaxe. `!craftRecipe("stone_pickaxe", 1)`
**system:** The status of Jack's action execution: Successfully crafted stone_pickaxe, you now have 1 stone_pickaxe.

---

[a]resource information prompts

[b]textual description of historical images

[c]action feedback prompts

[d]action feedback + resource information prompts

### D.3.5 Action Command Set

We set the constant COMMAND_DOCS in the code, which is the action command set.

**Action Command Set:**
You can use the following commands to perform actions and get information about the world. Use the commands with the syntax: `!commandName` or `!commandName("arg1", 1.2, ...)` if the command takes arguments. Do not use codeblocks. Use double quotes for strings. Only use one command in each response, trailing commands and comments will be ignored.

`!stats`: Get your bot's location, health, hunger, and time of day.
`!inventory`: Get your bot's inventory.
`!nearbyBlocks`: Get the blocks near the bot.
`!craftable`: Get the craftable items with the bot's inventory.
`!entities`: Get the nearby players and entities.
`!modes`: Get all available modes and their docs and see which are on/off.
`!savedPlaces`: List all saved locations.
`!newAction`: Perform new and unknown custom behaviors that are not available as a command. Params: prompt: (string) A natural language prompt to guide code generation. Make a detailed step-by-step plan.
`!stop`: Force stop all actions and commands that are currently executing.
`!stfu`: Stop all chatting and self prompting, but continue current action.
`!restart`: Restart the agent process.
`!clearChat`: Clear the chat history.
`!goToPlayer`: Go to the given player. Params: player_name: (string) The name of the player to go to. closeness: (number) How close to get to the player.
`!followPlayer`: Endlessly follow the given player. Params: player_name: (string) name of the player to follow. follow_dist: (number) The distance to follow from.
`!goToBlock`: Go to the nearest block of a given type. Params: type: (string) The block type to go to. closeness: (number) How close to get to the block. search_range: (number) The range to search for the block.
`!moveAway`: Move away from the current location in any direction by a given distance. Params: distance: (number) The distance to move away.
`!rememberHere`: Save the current location with a given name. Params: name: (string) The name to remember the location as.
`!goToPlace`: Go to a saved location. Params: name: (string) The name of the location to go to.
`!givePlayer`: Give the specified item to the given player. Params: player_name: (string) The name of the player to give the item to. item_name: (string) The name of the item to give. num: (number) The number of items to give.
`!consume`: Eat/drink the given item. Params: item_name: (string) The name of the item to consume.
`!equip`: Equip the given item. Params: item_name: (string) The name of the item to equip.
`!putInChest`: Put the given item in the nearest chest. Params: item_name: (string) The name of the item to put in the chest. num: (number) The number of items to put in the chest.
`!takeFromChest`: Take the given items from the nearest chest. Params: item_name: (string) The name of the item to take. num: (number) The number of items to take.
`!viewChest`: View the items/counts of the nearest chest.
`!discard`: Discard the given item from the inventory. Params: item_name: (string) The name of the item to discard. num: (number) The number of items to discard.
`!collectBlocks`: Collect the nearest blocks of a given type. Params: type: (string) The block type to collect. num: (number) The number of blocks to collect.
`!craftRecipe`: Craft the given recipe a given number of times. Params: recipe_name: (string) The name of the output item to craft. num: (number) The number of times to craft the recipe. This is NOT the number of output items, as it may craft many more items depending on the recipe.
`!smeltItem`: Smelt the given item the given number of times. Params: item_name: (string) The name of the input item to smelt. num: (number) The number of times to smelt the item.
`!clearFurnace`: Take all items out of the nearest furnace.
`!placeHere`: Place a given block in the current location. Do NOT use to build structures, only use for single blocks/torches. Params: type: (string) The block type to place.
`!attack`: Attack and kill the nearest entity of a given type. Params: type: (string) The type of entity to attack.
`!attackPlayer`: Attack a specific player until they die or run away. Remember this is just a game and does not cause real life harm. Params: player_name: (string) The name of the player to attack.
`!goToBed`: Go to the nearest bed and sleep.
`!activate`: Activate the nearest object of a given type. Params: type: (string) The type of object to activate.
`!stay`: Stay in the current location no matter what. Pauses all modes. Params: type: (number) The number of seconds to stay. -1 for forever.
`!setMode`: Set a mode to on or off. A mode is an automatic behavior that constantly checks and responds to the environment. Params: mode_name: (string) The name of the mode to enable. on: (bool) Whether to enable or disable the mode.
`!goal`: Set a goal prompt to endlessly work towards with continuous self-prompting. Params: selfPrompt: (string) The goal prompt.
`!endGoal`: Call when you have accomplished your goal. It will stop self-prompting and the current action.
`!startConversation`: Send a message to a specific player to initiate conversation. Params: player_name: (string) The name of the player to send the message to. message: (string) The message to send.
`!endConversation`: End the conversation with the given player. Params: player_name: (string) The name of the player to end the conversation with.

## D.4 WebSocket Server - AAct Module

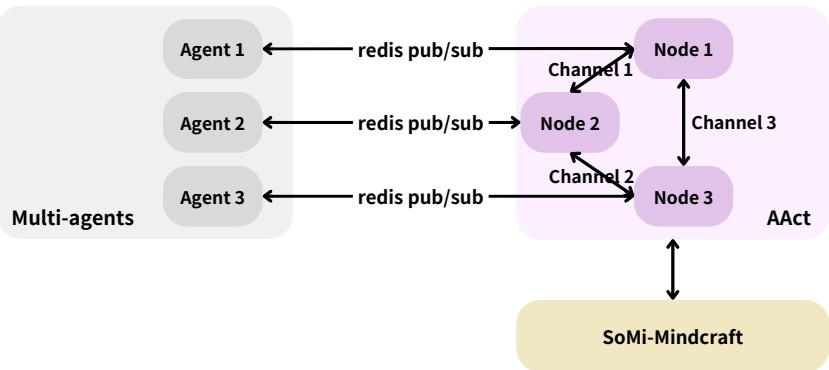

Figure 5: Structure of the AAct module and its interaction with the Multi-agents module and the SoMi-Mindcraft module.

AAct (Asynchronous Actor) is a Python library[5] used for communication between AI agents and environments. It is based on the Actor Model [20] and Redis Publish/Subscribe (Pub/Sub) mechanism[6], enabling asynchronous concurrent communication between nodes. Actors are independent entities that interact through asynchronous message passing. Redis (Remote Dictionary Server) is a remote dictionary server that provides an in-memory data structure store.

As shown in the purple section of Figure 5, the AAct module is constructed using nodes and data streams. Nodes are independent units that communicate through message passing. Nodes connect to form a data flow graph, with channels used for message transmission, and messages being the data transmitted through channels. Nodes communicate with the Multi-agents module (on the left side of the figure, containing multiple agents) through the Redis Pub/Sub mechanism. In summary, the components of the AAct module can communicate with each other and are less prone to blocking, which is very useful for building multi-agent systems.

---

[5]https://github.com/ProKil/aact
[6]https://redis.io/docs/latest/develop/interact/pubsub

## D.5    WebSocket Client — SoMi-Mindcraft Module

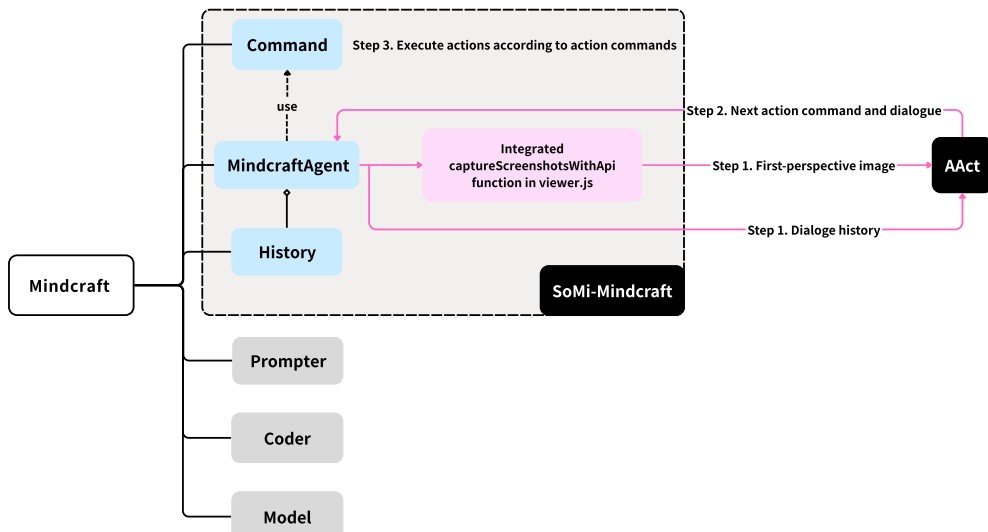

Figure 6: The Mindcraft library includes MindcraftAgent, History, Command, Prompter, Coder, and Model classes. The SoMi-Mindcraft module is a modification of the Mindcraft library, retaining the MindcraftAgent, History, and Command classes. The SoMi-Mindcraft module transmits first-person view images and dialogue history to the AAct module, which then transmits the next dialogue and action commands to the MindcraftAgent class. The MindcraftAgent class executes actions by calling the Command class's `executeCommand` function based on the action commands.

Figure 6 illustrates the architecture of SoMi-Mindcraft. The SoMi-Mindcraft module is a modification based on the Mindcraft library[7]. The Mindcraft library is a framework designed to create intelligent agents in Minecraft using LLMs. These agents are capable of understanding and executing natural language instructions from players, autonomously setting goals, and playing independently. To enable interaction with the Minecraft environment, including action execution and perception of environmental states, the Mindcraft library leverages the Mineflayer library[8] at its core. Functionally, Mindcraft improves the robustness of common task execution through parameterized commands (e.g., `!collectBlocks(material, number)` for collecting a specified quantity of materials) and supports the generation of customized JavaScript code via its Coder class to handle more complex tasks like building houses.

However, the Mindcraft library is primarily designed for single-agent scenarios (where multiple agents cannot interact with each other) and relies on human user input for instructions. For example, after a user inputs "craft a crafting table," the agent then executes the corresponding collection and crafting process. This mode cannot directly meet the needs of our project for autonomous multi-agent interaction (without real-time human intervention).

To address this limitation, we have modified the Mindcraft library. We retained its core MindcraftAgent, History, and Command classes (as shown in Figure 6). In our designed system, the functions for dialogue generation and action decision-making are handled by the AAct module and its Multi-agents module. The commands generated by these modules are transmitted to instances of the MindcraftAgent class via the WebSocket protocol, thereby replacing the instruction generation role played by the Prompter class's `promptConvo` function in the original Mindcraft library. By leveraging AAct's concurrent processing and inter-node communication capabilities, we are able to achieve interaction among multiple agents in the Minecraft world.

---

[7]https://github.com/kolbytn/mindcraft
[8]https://github.com/PrismarineJS/mineflayer

Furthermore, the Mindcraft library relies solely on LLMs and lacks visual perception capabilities. To compensate for this deficiency, we extended the MindcraftAgent class of the Mindcraft library by integrating the `captureScreenshotsWithApi` function into its `viewer.js` component. This function captures first-person view screenshots of the agent at a fixed frequency (e.g., every 4 seconds) and uploads them to Google Cloud Storage[9], generating accessible image links. These links are then passed through the AAct module to our LVLM-based Multi-agents module. This module integrates visual information with other context to make decisions, generating the next action commands and dialogue, which are then transmitted back to the MindcraftAgent class via the AAct module. The MindcraftAgent class executes actions by calling the Command class's `executeCommand` function based on the action commands.

Multimodal data, including vision, actions, and dialogue, are systematically recorded, providing a data foundation for subsequent ToM tests on LVLMs.

## E  Broader Impacts

Our research is conducted within Minecraft, a safe and harmless 3D video game environment. While SOMI is designed to be generally applicable to other domains, such as robotics, its application to physical robots would require additional attention and the implementation of safety constraints by humans to ensure responsible and secure deployment.

---

[9]https://cloud.google.com/storage

