# OpenReview forum: "SoMi-ToM: Evaluating Multi-Perspective Theory of Mind in Embodied Social Interactions"
_NeurIPS.cc/2025/Datasets_and_Benchmarks_Track — NeurIPS 2025 Datasets and Benchmarks Track poster_

### Official Review · Reviewer_F9vH · 2025-06-14

**Rating:** 5
**Confidence:** 4

**Summary:**

1. SOMI-TOM is a novel benchmark designed to evaluate the ability of AI agents to understand the mental states of others in complex, embodied social interactions within the Minecraft game environment.
2. It addresses the limitations of existing benchmarks by incorporating multi-perspective data (first-person perspective during interaction, third-person perspective after interaction) and evaluating ToM through multi-modal information (text, images, video).
3. The study compares human performance with state-of-the-art large vision-language models (LVLMs) and finds a significant gap between the two, highlighting the need for further improvement in AI agents’ ToM capabilities.

**Dataset Code Accessibility:**

Yes

**Ethical Considerations:**

No, there are no or only very minor ethics concerns

**Final Justification:**

The authors have addressed my concerns.

**Limitations Weaknesses:**

1. Lack of Environmental Complexity: The current version of SOMI-TOM only considers simple crafting tasks and does not yet explore more complex real-world environments with diverse interactions and challenges.
2. Assumptions about Social Dynamics: The benchmark assumes a unidirectional obstruction mode, which may not fully capture the complexity of real-world social dynamics and interactions.

**Strengths Contributions:**

1. Novel Benchmark: SOMI-TOM fills a gap in the literature by providing a comprehensive evaluation of multi-perspective ToM in embodied multi-agent interactions.
2. Multimodal Data: The benchmark utilizes rich multimodal data, including text, images, and video, providing a more realistic assessment of AI agents’ understanding of complex social situations.
3. Systematic Evaluation: The study compares human performance with LVLMs, providing valuable insights into the current state of AI agents’ ToM capabilities and identifying areas for improvement.

---

> ### Author Rebuttal · Authors · 2025-07-29
>
> Thank you for your positive review and recognition of our dataset's novelty and contribution in establishing a comprehensive benchmark for evaluating multi-perspective Theory of Mind (ToM) within complex, embodied social interactions. We appreciate your detailed feedback and will address your concerns as follows.
>
> **1. Lack of Environmental Complexity & 2. Assumptions about Social Dynamics**
>
> We thank the reviewers for their valuable feedback and acknowledge that the current version of the benchmark is simplified in terms of environmental complexity and social dynamics. This was a deliberate design choice to accommodate the capabilities of current LVLMs and to ensure that we can reliably evaluate their ToM.
>
> Our preliminary research found that for more complex tasks (such as cooking, building a house, obtaining rare materials and tools, etc.), current models have a lower success rate and are prone to getting agents stuck in loops of repetitive actions. This observation is consistent with the findings of existing research [1, 2], which indicate that the capabilities of current LVLMs are still limited in handling such tasks. Introducing tasks that are too difficult at this stage would introduce excessive noise, making it difficult for us to effectively evaluate the agent's ToM.
>
> Nevertheless, we fully agree with the reviewer's point of view. A good benchmark must continuously evolve to push the frontiers of the field and prepare for more powerful future models. Our SOMI environment is designed for this purpose with a high degree of scalability. We plan to enhance the benchmark in future work. As a first step, we will begin to integrate a select few more challenging tasks. In our longer-term plan, we will: (1) systematically increase the environmental complexity, introducing tasks that require longer interaction sequences and more diverse problem-solving strategies; and (2) explore more complex and multi-directional social dynamics, such as cooperative and competitive scenarios with richer background settings.
>
> [1] White, Isadora, et al. "Collaborating Action by Action: A Multi-agent LLM Framework for Embodied Reasoning." arXiv preprint arXiv:2504.17950 (2025).
>
> [2] Wang, Guanzhi, et al. "Voyager: An Open-Ended Embodied Agent with Large Language Models." Transactions on Machine Learning Research."

---

> > ### Comment · Area_Chair_pEW4 · 2025-08-06
> > **Post-rebuttal discussion**
> >
> > @F9vH
> >
> > Dear reviewer, please respond to the authors' rebuttal and explain what concerns are solved and what remain unaddressed.

---

> > ### Comment · Reviewer_F9vH · 2025-08-07
> >
> > Thanks for you rebuttal, you have addressed my concerns

---

> > > ### Author Response · Authors · 2025-08-08
> > >
> > > Thank you for your reply! We are pleased that we could address your concerns.

---

### Official Review · Reviewer_BGxz · 2025-07-01

**Rating:** 4
**Confidence:** 3

**Summary:**

The submission introduces SoMi-ToM, a novel benchmark for evaluating multi-perspective Theory of Mind (ToM) in embodied, multi-agent Minecraft interactions. Key contributions are: 1) the SoMi environment for generating multimodal interaction data; 2) the SoMi-ToM benchmark with first-person (real-time state inference) and third-person (post-task goal/behavior inference) evaluation; 3) a new dataset (35 videos, 363 images, 1225 questions); and 4) an LVLM vs. human evaluation showing a significant performance gap, indicating areas for LVLM improvement.

**Dataset Code Accessibility:**

Yes

**Ethical Considerations:**

No, there are no or only very minor ethics concerns

**Limitations Weaknesses:**

Limitation Weakness: 1. More concretely demonstrate SoMi-ToM's unique resolution of critical Theory of Mind (ToM) evaluation bottlenecks unaddressed by prior work, potentially through quantitative data comparisons beyond existing qualitative ones. 2. Deepen the dataset and benchmark contribution by better detailing technical challenges and solutions in its creation and offering richer statistical analysis from the collected data, such as how varying social dynamics within SoMi-ToM impact LVLM failure modes. 3. Urgently rectify critical dataset issues identified in the "Dataset Reviewer Report" by adding a clear open-source license (the report indicates "License: None" and flags "Missing: license" in core metadata ), ensuring all data files are accessible (the report flags "Inaccessible: 2 files" ), and significantly improving Croissant metadata (the report shows "Description Coverage: 0/20 (0%)" ); additionally, enhance Responsible AI (RAI) documentation (all RAI fields are reported as "Missing" ).

**Strengths Contributions:**

The SoMi-ToM benchmark is significant and novel, addressing the gap in existing Theory of Mind (ToM) evaluations by introducing a multi-perspective (first and third-person) approach in a dynamic, multi-agent Minecraft environment with social interactions. This work is impactful and relevant, as it highlights that current state-of-the-art Large Vision-Language Models (LVLMs) significantly underperform humans (with accuracy gaps of 40.1% in first-person and 26.4% in third-person evaluations), driving future research. The paper clearly distinguishes itself from prior work by comparing SoMi-ToM to existing benchmarks and justifying its necessity. Finally, the presentation is clear, well-organized, and easy to understand, with informative figures and tables effectively demonstrating the LVLM-human performance gap.

---

> ### Author Rebuttal · Authors · 2025-07-29
>
> Thank you for your positive review and recognition of our dataset's novelty, significance, and impact as an embodied benchmark for Theory of Mind (ToM). We appreciate your detailed feedback and will address your concerns as follows.
>
> **1. More concretely demonstrate SoMi-ToM's unique resolution of critical ToM evaluation bottlenecks unaddressed by prior work, potentially through quantitative data comparisons beyond existing qualitative ones.**
>
> In Appendix C, Table 5 (page 14), we compare and contrast our work with 19 previous ToM benchmarks, including quantitative comparisons of the "Number of Agents" and "Test Size," showing the novelty and uniqueness of our benchmark. In addition to the comparisons already present in the original paper, we have also added a comparison of the duration of individual videos for several video-based ToM benchmarks. The table is as follows:
>
> | Benchmark | Single Video Length |
> | :--- | :--- |
> | Phase | 10-25s |
> | Agent | 5.6-25.2s |
> | MMToM-QA | 1462 frames (average) |
> | Infant Cognition Benchmark | 20 frames |
> | MuMA-ToM | 36s (average) |
> | Amber van Groenestijn Benchmark | 271.82s (average) |
> | SoMi-ToM (Ours) | 263.14s (average) |
>
> Additionally, some benchmarks are purely text-based, for which a quantitative comparison would not be very meaningful. If there are other dimensions along which we should compare to prior work, please let us know!
>
> **2. Deepen the dataset and benchmark contribution by better detailing technical challenges and solutions in its creation and offering richer statistical analysis from the collected data, such as how varying social dynamics within SoMi-ToM impact LVLM failure modes.**
>
> Thank you for this suggestion; due to length limits, we predominantly elaborated on the technical challenges and our corresponding strategies in Appendix D.5 (and in the footnote on page four). For camera ready, we will better discuss the following in the main text:
>
> Challenge 1 (Appendix D.5, lines 579-601): We found that existing Minecraft interaction frameworks (such as Mindcraft) are primarily designed for single agents and rely on real-time command input from human users. This cannot meet our research needs for "multi-agent autonomous social interaction without human intervention."
> Solution: We designed and implemented a three-tiered asynchronous communication architecture consisting of a Multi-agents module, AAct middleware, and a SoMi-Mindcraft module. As shown in Appendix Figures 4 and 5, this architecture utilizes the publish/subscribe mechanism of the AAct module to achieve decoupling and efficient asynchronous message passing between multiple LVLM agents. This replaces the command generation role of the original framework, successfully enabling autonomous decision-making and continuous interaction among the agents.
>
> Challenge 2 (Appendix D.5, lines 602-611): The original framework lacked visual perception capabilities, preventing the agents from "seeing" the environment. This was inconsistent with our goal of evaluating Embodied ToM.
> Solution: We extended the MindcraftAgent class by integrating the `captureScreenshotsWithApi` function into the `viewer.js` component. This allows the agents to capture first-person perspective screenshots at a fixed frequency (every 4 seconds). This visual information is transmitted to the LVLM through our architecture, enabling its decision-making process to incorporate a true multimodal input of vision, dialogue, and game state history.
>
> Challenge 3 (Footnote on page 4): Generating data for benchmarking in an open world requires ensuring data stability and reproducibility amidst complex interactions.
> Solution: We guarantee that each task has the same environmental configuration and resource distribution by setting a fixed random seed for the world generator. Concurrently, the agents' spawn points retain a degree of randomness to increase the diversity of interactions.
>
>
> For the Chain-of-Thought method, we conducted a more detailed statistical analysis to study the **impact of social dynamics on the failure modes of LVLMs**. The results show that the impact is minor.
>
> *State Inference Failures*
>
> | Model | Failure Mode | Collaboration (Total: 630) | Obstruction (Total: 420) |
> | :--- | :--- | :---: | :---: |
> | **Gemini 1.5 Pro** | (1) Ignoring resource consumption | 101 / 282 | 64 / 169 |
> | | (2) Insufficient reliance on feedback | 83 / 282 | 37 / 169 |
> | | (3) Misled by initial intentions | 98 / 282 | 68 / 169 |
> | **Gemini 2.0 Flash** | (1) Ignoring resource consumption | 99 / 298 | 75 / 176 |
> | | (2) Insufficient reliance on feedback | 77 / 298 | 59 / 176 |
> | | (3) Misled by initial intentions | 122 / 298 | 42 / 176 |
> | **GPT-4o** | (1) Ignoring resource consumption | 89 / 268 | 61 / 157 |
> | | (2) Insufficient reliance on feedback | 65 / 268 | 37 / 157 |
> | | (3) Misled by initial intentions | 114 / 268 | 59 / 157 |
> | **InternVL2.5 78B**| (1) Ignoring resource consumption | 140 / 359 | 82 / 223 |
> | | (2) Insufficient reliance on feedback | 113 / 359 | 66 / 223 |
> | | (3) Misled by initial intentions | 106 / 359 | 75 / 223 |
> | **Qwen2.5-VL** | (1) Ignoring resource consumption | 131 / 348 | 52 / 164 |
> | | (2) Insufficient reliance on feedback | 102 / 348 | 66 / 164 |
> | | (3) Misled by initial intentions | 115 / 348 | 46 / 164 |
> | **LLaVA 1.6 13B** | (1) Ignoring resource consumption | 153 / 400 | 128 / 327 |
> | | (2) Insufficient reliance on feedback | 125 / 400 | 93 / 327 |
> | | (3) Misled by initial intentions | 122 / 400 | 106 / 327 |
>
> ---
>
> *Goal Inference Failures*
>
> | Model | Failure Mode | Collaboration (Total: 20) | Obstruction (Total: 15) |
> | :--- | :--- | :---: | :---: |
> | **Gemini 1.5 Pro** | (1) Overgeneralization | 0 / 0 | 1 / 1 |
> | | (2) Failure to identify hierarchical goals | 0 / 0 | 0 / 1 |
> | **Gemini 2.0 Flash** | (1) Overgeneralization | 0 / 0 | 0 / 0 |
> | | (2) Failure to identify hierarchical goals | 0 / 0 | 0 / 0 |
> | **GPT-4o** | (1) Overgeneralization | 1 / 2 | 0 / 1 |
> | | (2) Failure to identify hierarchical goals | 1 / 2 | 1 / 1 |
> | **InternVL2.5 78B**| (1) Overgeneralization | 1 / 2 | 1 / 3 |
> | | (2) Failure to identify hierarchical goals | 1 / 2 | 2 / 3 |
> | **Qwen2.5-VL** | (1) Overgeneralization | 0 / 0 | 0 / 1 |
> | | (2) Failure to identify hierarchical goals | 0 / 0 | 1 / 1 |
> | **VideoLLaMA 3 7B**| (1) Overgeneralization | 5 / 7 | 3 / 8 |
> | | (2) Failure to identify hierarchical goals | 2 / 7 | 5 / 8 |
> | **LLaVA-Video 7B**| (1) Overgeneralization | 2 / 4 | 2 / 5 |
> | | (2) Failure to identify hierarchical goals | 2 / 4 | 3 / 5 |
>
> ---
>
> *Behavior Inference Failures*
>
> | Model | Failure Mode | Collaboration (Total: 80) | Obstruction (Total: 60) |
> | :--- | :--- | :---: | :---: |
> | **Gemini 1.5 Pro** | (1) Entity recognition confusion | 8 / 24 | 7 / 25 |
> | | (2) Detail errors | 16 / 24 | 18 / 25 |
> | **Gemini 2.0 Flash**| (1) Entity recognition confusion | 6 / 16 | 5 / 19 |
> | | (2) Detail errors | 10 / 16 | 14 / 19 |
> | **GPT-4o** | (1) Entity recognition confusion | 7 / 15 | 3 / 12 |
> | | (2) Detail errors | 8 / 15 | 9 / 12 |
> | **InternVL2.5 78B**| (1) Entity recognition confusion | 8 / 20 | 8 / 24 |
> | | (2) Detail errors | 12 / 20 | 16 / 24 |
> | **Qwen2.5-VL** | (1) Entity recognition confusion | 5 / 15 | 5 / 18 |
> | | (2) Detail errors | 10 / 15 | 13 / 18 |
> | **VideoLLaMA 3 7B**| (1) Entity recognition confusion | 20 / 57 | 16 / 40 |
> | | (2) Detail errors | 37 / 57 | 24 / 40 |
> | **LLaVA-Video 7B**| (1) Entity recognition confusion | 12 / 44 | 13 / 37 |
> | | (2) Detail errors | 32 / 44 | 24 / 37 |
>
> **3. Urgently rectify critical dataset issues identified in the "Dataset Reviewer Report"**
>
> Thank you for your suggestion. We will improve the dataset accordingly.

---

> ### Comment · Area_Chair_pEW4 · 2025-08-06
> **Post-rebuttal discussion**
>
> @BGxz
>
> Dear reviewer, please respond to the authors' rebuttal and explain what concerns are solved and what remain unaddressed.

---

> ### Comment · Reviewer_BGxz · 2025-08-09
> **I will maintain my positive score**
>
> Thanks for the author's rebuttal. My concerns have been largely resolved, and I will maintain my positive score.

---

### Official Review · Reviewer_Uw9p · 2025-07-03

**Rating:** 4
**Confidence:** 2

**Summary:**

The paper presents SOMI-TOM, a Theory of Mind benchmark designed for evaluating embodied AI agents in complex, multi-agent social interactions within the Minecraft-based environment. The benchmark features multi-perspective, multimodal, and multi-stage multiple-choice questions based on agents completing diverse tasks. It combines first-person screenshots for real-time inference and third-person subtitled videos for post-task goal and behavior inference. A systematic evaluation reveals a performance gap between humans and sota LVLMs. Also, the paper show some qualitative analysis, highlighting common failures.

**Dataset Code Accessibility:**

Yes

**Ethical Considerations:**

No, there are no or only very minor ethics concerns

**Final Justification:**

Thank you to the authors for their detailed and thoughtful responses. My concerns have been addressed. I intend to maintain my original score.

**Limitations Weaknesses:**

1. Egocentric Input Format: Why does the egocentric view use only screenshots, while the exocentric view includes full videos? It would be helpful to explain this design choice.

2. Figure Readability: The system setting figures are text-heavy and difficult to read. Consider using color coding to visually separate different types of text and improve clarity.

3. Benchmark Comparison Table: It would strengthen the paper to include a comparison table showing how SOMI-TOM differs from previous ToM benchmarks across key dimensions—e.g., support for multi-view, multi-modal inputs, number of tasks/questions, and interaction complexity.

4. CoT Performance Justification: The paper notes that CoT only improves performance for Gemini 2.0 Flash, GPT-4o, and Qwen2.5-VL. More discussion on why these models benefit while others do not would add valuable insight.

**Strengths Contributions:**

1. The benchmark provides a new framework for studying ToM in open-ended, embodied settings and sets a foundation for future research in socially intelligent agents.

2. Unlike earlier Theory of Mind benchmarks that only look at single agents or simple settings, the proposed SOMI-TOM includes both first-person and third-person views. This makes it easier to see how well a model understands others during team tasks like helping or blocking.

3. The proposed SOMI-TOM uses Minecraft to create a rich environment where agents use text, visuals, actions, and game feedback to work together. Compared to prior work, this setup better reflects real-life situations.

---

> ### Author Rebuttal · Authors · 2025-07-29
>
> Thank you for your positive review and recognition of our dataset's novelty as a benchmark for embodied Theory of Mind (ToM), and its contributions through a rich, multi-agent environment and an innovative multi-perspective evaluation framework. We appreciate your detailed feedback and will address your concerns as follows.
>
>
> **1. Egocentric Input Format**
>
> The screenshot sequence we use (one frame every 4 seconds) is essentially a low-frame-rate video. Considering the agent's field of view does not change frequently, this sampling rate is sufficient to capture the key information for the task. We provide different forms of "video" input for the first-person and third-person perspectives based on the core differences in their evaluation objectives.
>
> For the first-person perspective, our goal is to evaluate real-time state inference. We want to know if the model can accurately infer the agent's beliefs (such as its inventory) at a specific moment based on previous events. The serialized screenshots allow us to precisely create these "point-in-time" test cases. If a full video were provided, it would be difficult to isolate and assess the model's "subjective instantaneous belief" at a particular moment.
>
> For the third-person perspective, the focus of the evaluation is on inferring the overall goal and behavior. This simulates an external observer who, after watching the entire process, understands the agent's final objective or overall actions. This naturally requires the full context provided by a complete video. From this viewpoint, the observer has no way of knowing the agent's exact inventory at every single moment, so a fine-grained, timestamped evaluation is not meaningful.
>
> **2. Figure Readability**
>
> Thank you for your suggestions. We will improve the readability of the figures in accordance with your feedback.
>
> **3. Benchmark Comparison Table**
>
> Thank you for that suggestion, and we had in fact already created this exact table  (Appendix C, Table 5, page 14). In that table, we compared our work with 19 previous benchmarks across several dimensions: Number of Agents, Perspective, Inter-agent Relationship, Concepts Tested, Test Size, Modality Communication, and Generation Evaluation. Please let us know if there are other comparison dimensions you would like to see.
>
> **4. CoT Performance Justification**
>
> The text (lines 278-282 and Table 3) mentions that in the Third-Person Perspective ToM Evaluation, GPT-4o showed the largest improvement of 13.2% after using the chain-of-thought (CoT) method. Gemini 2.0 Flash (1.1%) and Qwen2.5-VL (2.3%) had slight improvements. Other models showed a decrease in performance within 4%.
>
> The analysis of the reasons is as follows: (1) Models that can benefit from CoT (such as GPT-4o), when prompted, will generate detailed and logically rigorous reasoning steps, which are crucial for solving complex problems. In contrast, the reasoning processes generated by other models are relatively brief or have logical leaps, failing to provide positive guidance and may even interfere with their judgment [1, 2]. (2) The reasoning ability of CoT [3] is highly dependent on the model's scale, architecture, and whether it has been specifically optimized. Models like GPT-4o [4] already possess the capability to effectively execute such complex instruction chains, while other models may not have yet reached this threshold.
>
> In short, a performance improvement can only be achieved when the model's intrinsic reasoning level matches the requirements of the CoT method. We will add this discussion to the revised version of the paper.
>
>
> [1] Shi, Haojun, et al. "Muma-tom: Multi-modal multi-agent theory of mind." Proceedings of the AAAI Conference on Artificial Intelligence. Vol. 39. No. 2. 2025.
>
> [2] Jin, Chuanyang, et al. "MMToM-QA: Multimodal Theory of Mind Question Answering." Proceedings of the 62nd Annual Meeting of the Association for Computational Linguistics (Volume 1: Long Papers). 2024.
>
> [3] Kojima, Takeshi, et al. "Large language models are zero-shot reasoners." Advances in neural information processing systems 35 (2022): 22199-22213.
>
> [4] Hurst, Aaron, et al. "Gpt-4o system card." arXiv preprint arXiv:2410.21276 (2024).

---

> > ### Comment · Reviewer_Uw9p · 2025-08-05
> >
> > Thank you to the authors for their detailed and thoughtful responses. All of my concerns have been satisfactorily addressed. I will be maintaining my original score.

---

> > > ### Author Response · Authors · 2025-08-08
> > >
> > > We appreciate your recognition of our work and the effort you put into the rebuttal!

---

### Decision · Program_Chairs · 2025-09-18

**Decision:**

Accept (poster)

**Comment:**

The paper received three reviews with 2x borderline accept and 1x accept ratings. The comments are mostly positive: the reviewers found this new benchmark novel, which differs from existing benchmarks in that it offers a multi-perspective (first and third-person) setting and a dynamic, multi-agent environment. Most concerns raised by the reviewers were related to lack of details on some benchmark settings and lack of discussions on some results. These concerns have been properly addressed during the rebuttal. Since all reviewers are positive about this work and there is no major issue, the AC recommends that the paper be accepted.